# Molecular Detection and Genotyping of *Theileria* spp. in Deer (Cervidae) in Korea

**DOI:** 10.3390/microorganisms11112740

**Published:** 2023-11-09

**Authors:** Chang Uk Chung, Haeseung Lee, Min-Goo Seo, Seung-Hun Lee, Kyoo-Tae Kim, Kaifa Nazim, Jung-Sun Song, Dong Hwa Bae, Man Hee Rhee, Oh-Deog Kwon, Dongmi Kwak

**Affiliations:** 1College of Veterinary Medicine, Kyungpook National University, Daegu 41566, Daegu, Republic of Korea; ccu73@korea.kr (C.U.C.); koreasmg@knu.ac.kr (M.-G.S.); kyootae@knu.ac.kr (K.-T.K.); rheemh@knu.ac.kr (M.H.R.); odkwon@knu.ac.kr (O.-D.K.); 2Wild Animal Rescue Center, Andong 36605, Gyeongbuk, Republic of Korea; 3Veterinary Epidemiology Division, Animal and Plant Quarantine Agency, Gimcheon 39660, Gyeongbuk, Republic of Korea; lhspppp@korea.kr; 4College of Veterinary Medicine, Chungbuk National University, Cheongju 28644, Chungbuk, Republic of Korea; dvmshlee@chungbuk.ac.kr; 5Department of Veterinary Parasitology, Khalsa College of Veterinary & Animal Sciences, Amritsar 143002, Punjab, India; kaifa.nazim@gmail.com; 6Department of Veterinary Nursing Science, Yeungjin University, Daegu 41527, Daegu, Republic of Korea; anasong@yju.ac.kr (J.-S.S.); vetdonna@yju.ac.kr (D.H.B.); 7Cardiovascular Research Institute, Kyungpook National University, Daegu 41944, Daegu, Republic of Korea

**Keywords:** *Theileria*, tick, genotyping, deer, Korean water deer, phylogeny, incidence

## Abstract

Major clinical symptoms of *Theileria* infection include fever, anemia, anorexia, jaundice, and decreased milk production. Although several studies have been conducted on tick-borne pathogens, including *Theileria* in Korea, only a few have focused on *Theileria* infection in deer, including the Korean water deer. Blood samples from 160 deer were collected and subjected to DNA extraction and polymerase chain reaction (PCR). Next, PCR-positive samples were sequenced and analyzed by constructing a phylogenetic tree. The results showed that the overall infection rate of *Theileria* was 8.1% (13/160). Infection rates of 100% were observed in the northern and southern regions. However, the study’s limitation was its small sample size, wherein five and one samples were analyzed from the northern and southern regions, respectively. The central region exhibited the lowest infection rate of 2.9% (4/140). Infection rates also differed based on seasons, with the highest (18.4%, 9/49) being observed in spring, followed by that in summer (8.9%, 4/45). However, no infection was observed during autumn and winter. A phylogenetic analysis indicated that the PCR-positive samples contained *Theileria luwenshuni*, which usually infects small ruminants, such as goats and sheep.

## 1. Introduction

*Theileria* (next to *Babesia* or like *Babesia*) belongs to the Piroplasmorida group and is one of the two primary genera, along with *Babesia*, responsible for affecting domestic and wild animals [1]. Piroplasmosis is generally caused by parasites belonging to the *Babesia* and *Theileria* genera that infect humans, livestock, and wild animals, and it causes economic losses to agriculture. It can particularly affect livestock, meat, milk production, and the global livestock industry, including trade [2,3]. Ticks transmit the pathogens or etiologic agents that cause piroplasmosis. Some tick species can transmit several species of piroplasm; moreover, some piroplasm species can be transmitted by various phylogenetically related tick species [2]. The life cycle of most ticks is as follows: The first stage is an egg, which subsequently develops into a larva, a nymph, and an adult male or female. Engorged females spawn in concealed places such as under stones, lumps of dirt, cracks in walls, and gaps in the ground and wood because, unlike other parasites, ticks cannot directly lay eggs on the host. Mature females suck the bodily fluids of the host and then drop to the ground to lay eggs. During their lifespan of two to three years, females lay thousands of eggs, usually in spring, before dying. Ticks are one of the most significant blood-sucking arthropods and the most critical parasites worldwide, affecting both humans and animals. They can transmit various pathogens, and tick bites significantly increase the risk of developing diseases [4,5]. While feeding on a host with a blood-borne infection, ticks can ingest the pathogens along with the host’s blood. These pathogens can then survive and multiply within the tick’s body. Consequently, ticks can transmit the acquired pathogen to new hosts while feeding on them.

*Theileria* and *Babesia* are protozoan parasites that infect erythrocytes in vertebrate animals, including mammals. Although they share some similarities in their life cycles, there are several key differences between these parasites. When ticks, a vector, bite their animal hosts, they secrete infectious sporozoites along with their saliva, thereby infecting erythrocytes [6,7]. Sporozoites develop in erythrocytes into trophozoites and form merozoites. Merozoites destroy infected erythrocytes and reinfect other erythrocytes [6]. Some merozoites develop into gametocytes, which can develop into gametes when fed by ticks. Gametes fuse together to form zygotes, which then develop into kinetes, a motile state that can leave from the intestine of vector ticks [6]. The subsequent stages represent the differences between *Babesia* and *Theileria* [7]. *Theileria* sporozoites do not directly infect erythrocytes; instead, they penetrate lymphocytes (or macrophages) and develop into schizonts. Merozoites released from these schizonts enter erythrocytes, where they grow into nonpigment-forming piroplasms, proliferate through budding into four daughter cells, and form tetrads, which are often in the shape of a Maltese cross. *Babesia* parasites do not form pigments within the parasitic cells, which distinguishes them from the genera *Plasmodium* and *Haemoproteus*. *Babesia* directly infect erythrocytes, but they first infect lymphocytes (or monocytes, depending upon the *Theileria* sp.) and develop into schizonts, which release the erythrocyte-infective form of the parasite.

The traditional classification of *Theileria* and *Babesia* spp. is based on differences in their morphology, host specificity, and mode of transmission by a tick vector [8]. Although piroplasms were distinguished based on differences in their life cycle (e.g., transstadial transmission only occurs in *Theileria*, and they mainly reproduce in the lymphocytes), this classification method had limitations as it led to ambiguous groupings. These limitations have been addressed through the development of molecular methods such as 18S rRNA gene analysis for the phylogenetic analysis of *Theileria* and *Babesia* [9]. Molecular phylogenetic research based on the 18S rRNA gene sequence has facilitated the determination of the lineage of piroplasmids based on DNA sequence homology analyses. This method furnishes comprehensive biological information about the species of *Babesia* and *Theileria* and provided a basis for the recognition of phenotypes [10].

*Theileria* is an obligate hemoprotozoan parasite and a tick-borne pathogen. It is usually transmitted by hard ticks and causes theileriosis, which affects productivity and causes severe mortality in livestock husbandry (especially in cattle and horses) [11,12]. There was relatively little interest in ticks or *Theileria* in ruminants; however, piroplasmosis and infection by *Theileria* in small ruminants such as goats and sheep has recently garnered considerable interest in many countries [8]. The genus *Theileria* comprises several species, including those that infect wildlife and exhibit host specificity. Many studies on theileriosis have reported the presence of *T. parva* and *T. annulata* in cattle, *T. ovis* and *T. lestoquardi* in small ruminants such as sheep and goats, *T. annae* in foxes, *T. equi* in horses, and *T. youngi* in woodrat (*Neotoma fuscipes*) [8,13,14,15,16].

*Theileria* spp. can be divided into transforming and nontransforming groups based on their ability to transform the environment of host leukocytes into a state suitable for the proliferation of the parasites. The transforming group includes *T. parva*, *T. annulata*, *T. taurotragi*, and *T. lestoquardi*, whereas the nontransforming group includes *T. mutans*, *T. cervi*, and *T. velifera*. Although species belonging to the nontransforming group cannot induce proliferation, they can damage livestock [17,18].

Possible clinical signs and pathological manifestations in animals infected with *Theileria* include fever, chronic anemia, anorexia, jaundice, pyrexia, decreased milk production, weight loss, lymph node swelling, and leukopenia [19,20,21]. Diagnosis is usually made based on clinical signs and microscopic examination. Microscopic examination requires lymph node biopsy. On Giemsa or Wright staining, macroschizonts can appear round, oval, or pear-shaped. Serological methods, such as the immunofluorescent antibody test, can also be used, but they are only effective for the identification of the infection in recovered animals. The most sensitive and specific diagnostic method is polymerase chain reaction (PCR) [22,23]. Treatment for theileriosis has not yet been established; however, buparvaquone is considered a promising drug [24].

Studies on the detection and identification of *Theileria* spp. have been conducted in many countries. In South Africa, a study was conducted on the detection of *Theileria* spp. in dogs [21] and domestic ruminants in Ethiopia [25]. A survey on piroplasmosis in wildlife (*Babesia* and *Theileria* affecting free-ranging ungulates and carnivores) was conducted in Italy [26], and another survey was conducted in Great Britain on the detection of *T. luwenshuni* in sheep [27]. Dutch and Chinese researchers have also studied the characteristics of *T. buffeli* and *T. orientalis* [23]. Recently, studies from North America (Canada, the USA, and Mexico) on babesiosis and theileriosis have reviewed the pathogenesis, diagnosis, and epidemiology in humans, livestock, and wild animals [28]. In Mexico, a report was published regarding the molecular detection of *T. equi* in horses [29]. In Spain, a study was conducted on the prevalence of *Theileria* in roe deer (*Capreolus capreolus*) [30]. In Brazil, the occurrence of *Theileria* spp. in brown brocket deer (*Mazama gouazoubira*) and marsh deer (*Blastocerus dichotomus*) has been reported [31]. A review was conducted in New Zealand on *Theileria* in cattle from a disease-monitoring and epidemiological perspective [32,33]. In Australia, a study on the molecular-based detection and characterization of *Babesia* and *Theileria* in hard ticks was conducted [34]. Studies from Asia have reported theileriosis in cattle and small ruminants in India and Pakistan and the detection of *T. luwenshuni* in goats in Myanmar [35,36,37].

Several studies have also been conducted in the neighboring countries of Korea, such as China and Japan. Phylogenetic analyses of the genetic diversity [18], prevalence, and molecular characterization of *Theileria* spp. in sika deer were conducted in Japan [38]. In China, studies on *Theileria* infection using molecular methods for the diagnosis of piroplasms (including *Theileria*) in sika deer [39] and on the prevalence of *Theileria* in goats, sheep, and cattle [40,41] have been conducted.

Several studies have been published on the infection of *Theileria* in Korea since the 20th century. Although most studies were restricted to cattle, some assessed the status of *Theileria* in ticks [11,42]. A study investigated *Theileria* spp. in ticks in Korea and evaluated the potential risk to wildlife and livestock [11]. In wildlife, studies have been conducted on tick-borne pathogens (including *Theileria*) in Korean water deer (*Hydropotes inermis argyropus*) [43,44,45]. The infection rate of *Theileria* in dairy cattle (Holstein cattle) based on the grazing period depending on the season [46] and the detection of *T. equi* through a phylogenetic analysis of equine piroplasms [16] have been studied.

Korean water deer are one of the most widely distributed wild animals in Korea and have been reported to serve as a significant reservoir for various tick-borne pathogens (including *Theileria*) that can infect livestock [43,44]. As only a few studies have focused on the infection of *Theileria* in Korean water deer with a limited number of samples [43,44], the infection pattern and genotypic characteristics of *Theileria* in deer including Korean water deer in a relatively larger sample size was investigated.

## 2. Materials and Methods

### 2.1. Blood Sample Collection

Blood samples of 160 deer were collected between December 2014 and February 2023 from 16 cities in Korea. All blood samples were taken from the bodies of deer that died in a road accident. All blood samples were collected in separate tubes and stored at 4 °C before DNA extraction. DNA was extracted within 1–2 days after sample collection.

Data regarding species, region, and season were recorded for each sample whenever possible. The study area was categorized into three groups according to the provincial boundaries: northern (Gangwon (GW) and Gyeonggi (GG) provinces), central (Chungbuk (CB), Chungnam (CN), and Gyeongbuk (GB) provinces), and southern (Jeonbuk (JB), Jeonnam (JN), Gyeongnam (GN), and Jeju (JJ) provinces) regions (Figure 1). Samples with unclear information were labeled “unknown”. Although individual information was labeled at the time of sample collection, there were cases where the label was removed or inaccurate information was recorded in the process of being transferred for autopsy and specimen collection. These cases were classified as unknown samples.

### 2.2. DNA Extraction and PCR Assay

DNA was extracted using the DNeasy^®^ Blood & Tissue Kit (Qiagen, Hilden, Germany) according to the manufacturer’s instructions. The quality and quantity of DNA were estimated using the Infinite^®^ 200 PRO NanoQuant plate reader (Tecan, Mannedorf, Switzerland). The extracted DNA was stored at −20 °C until it was subjected to PCR.

The PCR premix was prepared using AccuPower^®^ HotStart PCR Premix (Bioneer, Daejon, Korea). The lyophilized premix comprised 1 μL of each PCR primer, 2 μL of DNA template, and 16 μL of distilled water. For the first round of PCR, which was performed to identify the samples positive for piroplasm, the piroplasm 18S rRNA-specific primers BT-F1 (5′-GGT TGA TCC TGC CAG TAG T-3′) and BT-R2 (5′-TTG CGA CCA TAC TCC CCC CA-3′) as well as BT-F3 (5′-TGG GGG GAG TAT GGT CGC AAG-3′) and BT-R3 (5′-CTC CTT CCT TTA AGT GAT AAG-3′) sets were used as previously described [16]. The PCR process included an initial denaturation step of 10 min at 94 °C; 30 cycles of 30 s at 95 °C, 30 s at 65 °C, and 45 s at 72 °C (for BT-F3/BT-R3 primers, 30 s at 61 °C and 30 s at 72 °C); and a post-elongation of 10 min at 72 °C. The expected amplicon sizes were 1024 and 644 bp, respectively. The next round of PCR was performed to distinguish 18S rRNA of *Theileria* and *Babesia*. Tsp-F (5′-GTT ATA AAT CGC AAG GAA GTT TAA GGC-3′) and Tsp-R (5′-GTG TAC AAA GGG CAG GGA CGT A-3′) were used as forward and reverse primers, respectively, to identify *Theileria* [46]. The steps of the PCR reaction included an initial denaturation step of 5 min at 95 °C; 40 cycles of 30 s at 95 °C, 30 s at 61 °C, and 30 s at 72 °C; and a post-elongation of 5 min at 72 °C. The expected amplicon size was 239 bp. Following the amplification of *Theileria* 18S rRNA, electrophoresis was performed on a 1% agarose gel for 30 min at 130 V, and ethidium bromide was used for staining. Gel Doc™ EZ Imager (Bio-Rad, Pleasanton, CA, USA) was used to obtain images and to confirm the positive bands.

### 2.3. Statistical and Phylogenetic Analysis

The χ2 test was performed using SPSS version 26.0 (IBM Corporation, Armonk, NY, USA) to test statistical significance. *p* values of <0.05 were considered statistically significant. Of the 160 blood samples, 13 PCR-positive samples were sent to the Macrogen co. (Daejeon, Korea) for bidirectional nucleotide sequencing. These were aligned with the other sequences of *Theileria* obtained from the GenBank database using BioEdit 7.2.5 and MEGA7 software (Pennsylvania State University, State College, PA, USA). The obtained sequences were compared with other sequences using NCBI Web BLAST (available at: https://blast.ncbi.nlm.nih.gov/Blast.cgi, accessed on 30 August 2023). A phylogenetic tree was constructed, and a phylogenetic analysis was performed using the maximum likelihood method with 1000 bootstrap replicates.

## 3. Results

### 3.1. Region-Wise Infection Rates of Theileria

Among the 160 deer blood samples, 1, 5, and 140 were collected from the southern, northern, and central regions, respectively; the origins of the remaining 14 samples were unknown. In comparison with the central region, which showed an infection rate of 2.9% (4/140), other regions exhibited higher infection rates. An infection rate of 100% was observed in the northern and southern regions despite the small sample size (Table 1). Infection rates significantly differed among the regions (*p* < 0.001).

### 3.2. Seasonal Variations in Theileria Infection Rates

The highest *Theileria* infection rate of 18.4% (9/49) was observed in spring, followed by summer, with an infection rate of 8.9% (4/45). No infection rate was detected in the remaining two seasons (Table 2). Infection rates significantly differed among seasons (*p* = 0.014).

### 3.3. Sequencing and Phylogenetic Analysis

In the first step, 13 positive samples were confirmed by performing PCR to screen piroplasm-positive samples, and sequencing of these samples yielded two sequences. Next, the sequencing of positive samples was performed to distinguish the 18S rRNA of *Babesia* and *Theileria*, which yielded only one sequence. All these sequences were submitted to GenBank (accession numbers OR458386, OR458387, and OR398798). These samples were used for the phylogenetic analysis. A phylogenetic tree was constructed using these sequences and other sequences chosen from the GenBank database. A phylogenetic tree was constructed based on the sequences obtained using piroplasm universal primers and those obtained from *Babesia* genera, such as *B. microti*, *B. gibsoni*, and *B. canis*, and *Theileria* genera, such as *T. sergenti*, *T. orientalis*, and *T. cervi*; these sequences were analyzed together. The obtained piroplasm sequences were confirmed to belong to *Theileria* (Figure 2). Sequences obtained using *Theileria*-specific primers were grouped along with *T. luwenshuni* in the phylogenetic tree and were identified as *T. luwenshuni*. Other sequences used for data analysis included *T. orientalis*, *T. cervi*, *T. annulata*, *T. parva*, *T. capreoli*, *T. buffeli*, and *T. sergenti* (Figure 3).

## 4. Discussion

Several studies on tick-borne diseases have been conducted worldwide because of their pathological and zoonotic significance [45,47]. In addition, climatic changes and decreased environment-related diversity have increased the infection rate of tick-borne diseases [43,48]. Piroplasmorida, which includes *Babesia* and *Theileria*, is a class containing major pathogens that cause tick-borne diseases [1]. Infection rates of *Babesia* and/or *Theileria* were 89.7% (156/174) in roe deer in Spain [30], 58.3% (7/12) in brown brocket deer (*Mazama gouazoubira*) and marsh deer (*Blastocerus dichotomus*) in Brazil [31], and 35.3% (24/68) in sika deer (*Cervus nippon*) in China [39]. In addition, the infection rate of theileriosis in other animals has been studied. The infection rates were 4.7% (12/257) in wild boar, 22.2% (8/36) in Alpine chamois, and 1.0% (2/205) in red foxes in Italy [26]; 80.0% (80/100) in cattle, 93.8% (150/160) in sheep, and 1.9% (5/265) in goat, with 235/525 (44.8%) of the ruminants testing positive for *Theileria* in Ethiopia [25]; and 6.2% (82/1329) in dogs in South Africa [21]. In Korea, as in other countries, ticks are ubiquitous. Several studies on tick-borne diseases have been conducted in cattle, dogs, and horses [43]. The rate of infection was 1.2% (9/737) in cattle [49], 3.1% (16/510) in dogs [50], and 0.9% (2/224) in horses [16].

In this study, DNA was extracted from the blood samples of 160 deer. After DNA extraction, PCR and electrophoresis were performed. Based on the obtained results, phylogenetic and statistical analyses were performed. The overall infection rate of *Theileria* in Korean water deer was 8.1% (13/160) in this study. The infection rates in Korean water deer from other studies varied from 80.0% (8/10) in the Jeonbuk Province [43] to 77.8% (14/18) in the southern areas [51] and 72.2% (13/18) in the Chungbuk Province [45]. However, *Theileria* infection was not detected in 28 Korean water deer in Jeonbuk Province [44]. This difference could be attributed to factors such as the experimental environment, sample size, season, or area of sample collection. Furthermore, a large-scale study is warranted because small sample sizes included in previous studies render their results unreliable.

To examine the effect of region on the infection rate, Korea was divided into three regions: northern, central, and southern. When information on a region was unclear or inadequate, the region was marked as “unknown”. The highest infection rate of 100% was observed in the northern and southern regions. The central and “unknown” regions demonstrated an infection rate of 2.9% (4/140) and 21.4% (3/14), respectively. Infection rates were significantly different among the studied regions (*p* = 0.001). Nevertheless, the validity of these results was limited because the samples were concentrated in the central region, especially the Gyeongbuk Province.

In terms of season, spring demonstrated the highest infection rate of 18.4% (9/49), followed by summer (8.9%, 4/45). The infection rates in all seasons were significantly different (*p* = 0.014). However, this comparison has limited validity owing to the limited number of samples collected in autumn and winter. The findings of this study were similar to those of previous studies conducted in Korea. For example, the previously reported infection rate was 5.1% (9/178) in summer [49], 4.9% (8/163) in summer [50], and 2.5% (5/202) in spring. Additionally, infection rates of 1.7% (2/120) in winter and spring and no infection rate in summer and autumn have also been recorded [51]. In New Zealand, the *Theileria* infection rate was higher in spring and autumn [32,33]. High infection rates of 19.6% (20/102) and 45.4% (45/99) were reported in summer and autumn in India, respectively [35]. The infection rate in the spring (8%, 8/63) and summer (19.6%, 18/92) seasons was higher than that in winter (2.1%, 1/48) in Pakistan [36]. Overall, higher infection rates were recorded in spring and summer than those in the other seasons, indicating that wet conditions facilitated *Theileria* infection. However, as no positive samples were collected during autumn and winter in this study, this aspect remains unclear.

To date, several species of *Theileria* have been confirmed to infect small ruminants, including *T. ovis*, *T. separate*, and *T. recondite*, along with *T. luwenshuni*, *T. lestoquardi*, and *T. uilenbergi*, which are highly pathogenic to sheep and goats [52]. The *Theileria* spp. identified in Korea include *T. luwenshuni*, *T. ovis*, *T. orientalis*, and *T. equi* [8,29,48,49]. *T. equi* has been confirmed in horses in Gyeonggi province (0.9%, 2/224) [16]; *T. orientalis* was identified in cattle and the ticks infesting them [49]; *T. luwenshuni* and *T. ovis* were reported in Korean water deer and deer ked (*Lipoptena fortisetosa*) [52]; and *T. luwenshuni* was identified in ticks collected from Siberian roe deer (*Capreolus pygargus*), native Korean goats (*Capra hircus coreanae*), and horses [53].

No species other than *T. luwenshuni* could be identified in this study. *T. luwenshuni* is one of the most common pathogenic species that infect small ruminants such as sheep, goats, and deer [11]. *T. luwenshuni* was highly prevalent in roe deer on Jeju Island and detected in the blood samples of all 23 deer (100%) and 8 tick pools (34.8%) [54]. In the mid- and southwestern regions of Korea, 541 tick pools were tested, of which 211 (39.0%) were positive for *Theileria* [11]. Interestingly, the ticks belonging to the *Haemaphysalis* spp. were found to be most infected with *T. luwenshuni*, with *H. longicornis* being the most commonly infected tick with an incidence rate of 92.0% (161/175), followed by *H. flava* at 5.1% (9/175) [11]. *H. longicornis* is the most commonly found tick in Korea and has been detected in Jeju Island, far from the mainland [53]. Therefore, it is a vital natural vector for *T. luwenshuni* [11]. Additionally, *H. longicornis* is a vector for tick-borne pathogens and can transmit diseases to humans and animals [53]. Although this study did not directly identify ticks infesting deer and tick-borne pathogens (*Theileria* spp.), they can be identified in the blood samples of wild deer. Therefore, further research is needed regarding ticks occurring in deer that are presumed to transmit this pathogen.

The phylogenetic tree of piroplasms confirmed that they can be broadly classified into two groups: the *Theileria* spp., and *Babesia* spp. groups. In the phylogenetic tree of *Theileria*, *B. microti* was set as an outgroup, and several different *Theileria* sequences were analyzed. Based on the two phylogenetic trees, it could be confirmed that the sequences obtained in this study may commonly be identified as those related to *T. luwenshuni,* which mainly infects goats, sheep, and deer [27,37]. As only a short length of sequence was obtained using 216 bp amplicon, it is necessary to analyze the full-length sequencing to identify species in the future. In this study, the two sequences identified in the phylogenetic tree of the piroplasms, OR458386 and OR458387, were 100% and 99.89% similar to the sequences identified in deer ked in Korea, respectively. The sequence identified in the *Theileria* phylogenetic tree (OR398798) showed 100% similarity with the sequences identified from deer (FJ599640), sheep (JX469518), and camel (KU554730), although it was shorter than the sequence of piroplasms.

## 5. Conclusions

The prevention of *Theileria* infection is gaining urgency because it infects most livestock such as goats, sheep, and cattle. The overall *Theileria* infection rate in this study was 8.1% (13/160). Significant differences in infection rates among the various regions and seasons (*p* < 0.001 and *p* = 0.014, respectively) were observed. Phylogenetic analysis showed that *Theileria*-positive blood samples contained *T. luwenshuni*, which mainly infects deer, sheep, and goats.

The primary importance of this study is that it is one of the few studies that focuses on *Theileria* infection in deer, including Korean water deer, with a large sample size. However, biased sample collection based on the region and shortage of classification standards due to characteristics of wild animals may compromise the validity of these results. Therefore, further studies focusing on livestock and other *Theileria* spp. are warranted. In addition, it is also necessary to perform an analysis of *Theileria* species in ticks infesting wild deer, which were not included in this study. Because enzootic stability/instability cannot be excluded in livestock theileriosis, differences in prevalence may appear in various ways. Therefore, additional comprehensive study considering the life cycle of *Theileria* should be conducted to confirm the prevalence between the host and the pathogen, to prevent the spread of tick-borne pathogens, and to prepare for the prevention of infections in wild animals and nearby livestock farms.

## Figures and Tables

**Figure 1 microorganisms-11-02740-f001:**
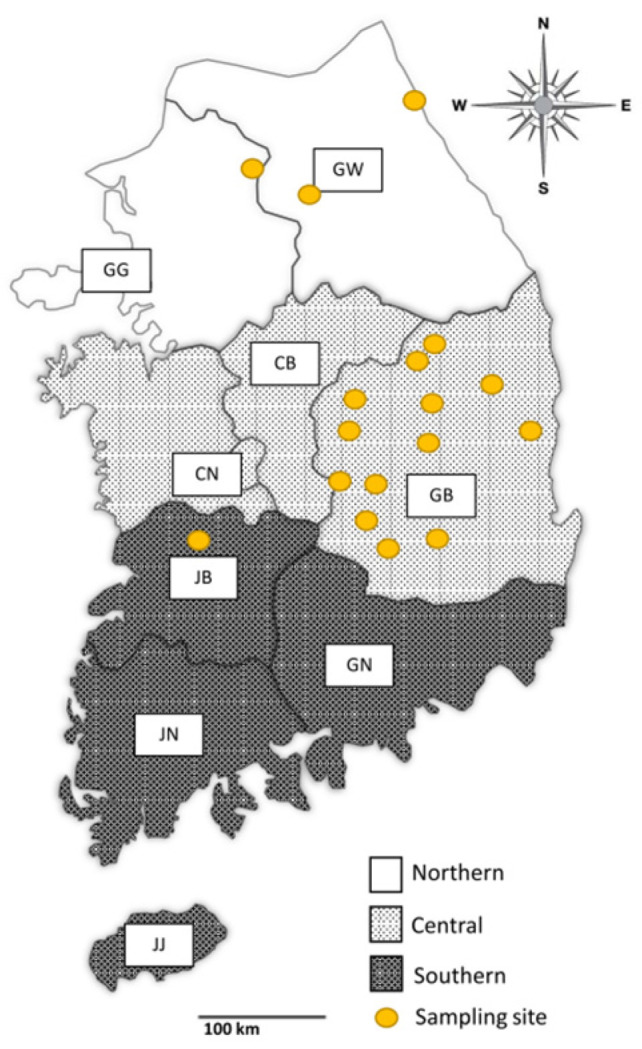
Map representing the regions from which samples were collected. Based on the provincial areas, the country was categorized into three regions: northern (Gangwon (GW) and Gyeonggi (GG) provinces), central (Chungbuk (CB), Chungnam (CN), and Gyeongbuk (GB) provinces), and southern (Jeonbuk (JB), Jeonnam (JN), Gyeongnam (GN), and Jeju (JJ) provinces) regions. Yellow spots indicate the sampling site.

**Figure 2 microorganisms-11-02740-f002:**
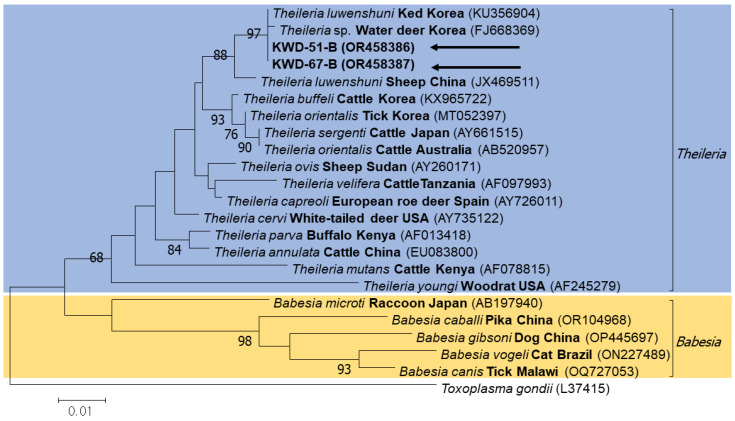
Genera corresponding to the piroplasm 18S rRNA (including *Theileria* and *Babesia*) were analyzed by constructing a phylogenetic tree, and the acquired sequences belonged to *Theileria*. The sequences obtained in this study are indicated using black arrows.

**Figure 3 microorganisms-11-02740-f003:**
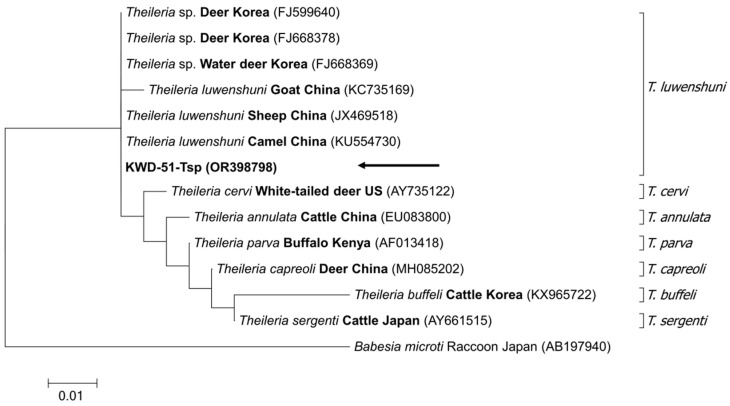
Phylogenetic tree of the *Theileria* 18S rRNA. The sequence obtained in this study is indicated using a black arrow.

**Table 1 microorganisms-11-02740-t001:** Region-wise infection rates of *Theileria*.

Region	No. Tested	No. Positive (%)	*p* Value
Northern	5	5 (100)	<0.001
Central	140	4 (2.9)	
Southern	1	1 (100)	
Unknown	14	3 (21.4)	
Total	160	13 (8.1)	

**Table 2 microorganisms-11-02740-t002:** *Theileria* infection rates based on season.

Season	No. Tested	No. Positive (%)	*p* Value
Spring	49	9 (18.4)	0.014
Summer	45	4 (8.9)	
Autumn	26	0	
Winter	33	0	
Unknown	7	0	
Total	160	13 (8.1)	

## Data Availability

Data supporting the conclusions of this article are included within the article. The newly generated sequences were submitted to the GenBank database under the accession numbers OR458386-OR458387 and OR398798. The datasets used and/or analyzed during the present study are available from the corresponding author upon reasonable request.

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
