# Peer review of "Molecular Detection and Genotyping of Theileria spp. in Deer (Cervidae) in Korea"

_microorganisms, 2023, doi:10.3390/microorganisms11112740_

Round 1

Reviewer 1 Report

Comments and Suggestions for Authors

The submitted manuscript describes a study focused on the infection pattern and genotypic characteristics of Theileria spp. in Korean water deer.

Several studies aimed to estimate the infection rates of Theileria spp. in both, domestic and wild hosts in Korea, have been published, including a few studies carried out to detect the infection of Theileria spp. in Korean water deer. As such, the manuscript presented is not original.

The novelty of this study, perhaps, relies on the approach used to gather the data. By combining the data obtained by PCR análisis on blood samples retrospectively collected from 160 deer between December 2014 to February 2023 from 16 cities in Korea, with a phylogenetic analysis performed with the obtained DNA sequences of several PCR amplicons It is shown that the Theileria-positive blood samples contained T. luwenshuni, an important piroplasmid parasite that infects mainly deer, sheep, and goats.

Thus, the results could be of clinical veterinary importance as it is shown that a substantial proportion of wáter deer (8.1%, representing 13 among the 160 deer blood samples analyzed in the study) was positive, indicating a potential reservoir for Theileria spp. which might be transmissible among the different species of mammals and ticks found in Korea. The results obtained are thus of primary interest to the local authorities and cattle farmers, but could also be of regional importance for the cattle industry in Southeast Asia, and provide with additional data that will contribute to the knowledge and understanding of the global distribution of tick-borne pathogens such as Theileria. Overall, the manuscript is well written, the laboratory PCR detection methods utilized are adequate, and the technical details are provided to replicate the work. However, although providing useful information, the paper should be improved before being accepted for publication.

1. Introduction

Line 42. It is stated that “Ticks transmit piroplasmosis, which act as vectors for infecting other animals”. Authors are requested to consider using instead “Ticks transmit the pathogens or etiologic agents that cause piroplasmosis…”

Line 46. Can use “Engorged females” instead of “They”.

Line 50. Can use “they turn themselves as ectoparasites…” instead of “parasites”

Line 59. The sentence “They can also transmit acquired diseases to new hosts when they feed” can be better read using “They can transmit the acquired pathogen to new hosts when they feed upon”.

Line 69. Please revise the statement “The latter stages represent the differences between Babesia and Theileria parasites”, and specify if the differences are in terms of morphology or biological function in the life cycle of each parasite genus.

Line 72. Revise sentence “whereas Theileria do not infect erythrocytes but enter lymphocytes and develop into schizonts.”. Can use “…but infect first the lymphocytes (or monocytes, depending upon the Theileria sp) and develop into schizonts, from which erythrocyte-infective forms are released.

Line 82. Please, complete the sentence “…plasmids based on homology.” Use “…based on DNA sequence homology analysis.”

Lines 90 and 98. Use “species” instead of “spp”

Lines 102. Authors should carefully check the statement “ When humans are affected with theileriosis…,”. Authors need to be careful about this statement. It has not been described so far that humans can be infected with Theileria spp. References 23 and 24, refer to human babesiosis.

Line 104.   The citation to reference number 22 is missing, it should go before these [23,24].

Lines 122-123. It is stated that “In Brazil, the occurrence of Theileria spp. in species of deer was reported [32,33].  Check reference 32. The paper refers to detection of Babesia and Theileria in roe deer in Spain.

Line 145. Rephrase statement “…(including Theileria) that can infect humans and livestock”. See the comment above on Theileria infecting humans….

2. Materials and Methods

Line 151. Authors are asked to describe the type of study carried out (purposive), how sample size was estimated, what the estimated water deer population in Korea, etc...

Line 193. Please indicate if all the small-size amplicons (Theileria sp) or any other from the larger Piroplasmid amplicon(s) were sent out for DNA sequencing

4. Discussion

Lines 272-278. Authors compare the results in terms of the infection rates obtained in their study carried out on water deer samples with “Previous studies on tick-borne diseases in Korea” and cite them in references [52] and [54]. However, these studies refer to canine pathogens (Anaplasma phagocytophilum, Hepatozoon canis, and Mycoplasma haemocanis) and cattle pathogen (Theileria sergenti) samples, respectively. Authors should indicate the relevance of such a comparison. Do water deer comingle with dogs and cattle? Do they share the same ticks as vectors??

Line 294. Use “species” instead of “spp”

Lines 295-296. Delete repeated species names “T. luwenshuni, T. lestoquardi, and T. uilenbergi”… [56]. Check for reference [55], It has not been cited up until this part of the discussion.

Line 304. Use “species” instead of “spp”

Lines 321-322. It is stated that “From the two phylogenetic trees, it could be confirmed that the sequences obtained in this study may commonly be identified as those related to T. luwenshuni, which mainly infect goats, sheep, and deer”. Authors should be careful on this statement, as the full length ssuRNA gene was not sequenced. A blast homology analysis performed by the reviewer, using the OR 398798, 216 bp sequence  and excluding T. luwenshuni from the search, showed 100% DNA sequence identity among the 216 bp amplicon and several Theileria spp. sequences, including Theileria ovis, Theileria mutans, and Theileria spp in dogs. 

Lines 328-330. The statement “These high similarities suggest that Theileria is potentially transmissible among the different species of mammals, ticks, and keds” should be revised. It could also mean that other Theileria spp are circulating in these hosts/vectors, not necessarily being T. luwenshuni.

Lines 345-347. Authors conclude that comprehensive studies should be conducted “to confirm the prevalence between the host and the pathogen, prevent the spread of tick-borne pathogens, and prepare to prevent infection in wild animals and nearby livestock farms.”. Authors should speculate a little bit on how this can be accomplished, having various scenarios with different prevalence rates. How about enzootic stability / instability issues on livestock theileriosis?

Comments on the Quality of English Language

To my opinion, the English language requires minor editing to understand the manuscript.

Author Response

Response to Reviewer 1

Comments and Suggestions for Authors

The submitted manuscript describes a study focused on the infection pattern and genotypic characteristics of Theileria spp. in Korean water deer.

Several studies aimed to estimate the infection rates of Theileria spp. in both, domestic and wild hosts in Korea, have been published, including a few studies carried out to detect the infection of Theileria spp. in Korean water deer. As such, the manuscript presented is not original.

The novelty of this study, perhaps, relies on the approach used to gather the data. By combining the data obtained by PCR análisis on blood samples retrospectively collected from 160 deer between December 2014 to February 2023 from 16 cities in Korea, with a phylogenetic analysis performed with the obtained DNA sequences of several PCR amplicons It is shown that the Theileria-positive blood samples contained T. luwenshuni, an important piroplasmid parasite that infects mainly deer, sheep, and goats.

Thus, the results could be of clinical veterinary importance as it is shown that a substantial proportion of wáter deer (8.1%, representing 13 among the 160 deer blood samples analyzed in the study) was positive, indicating a potential reservoir for Theileria spp. which might be transmissible among the different species of mammals and ticks found in Korea. The results obtained are thus of primary interest to the local authorities and cattle farmers, but could also be of regional importance for the cattle industry in Southeast Asia, and provide with additional data that will contribute to the knowledge and understanding of the global distribution of tick-borne pathogens such as Theileria. Overall, the manuscript is well written, the laboratory PCR detection methods utilized are adequate, and the technical details are provided to replicate the work. However, although providing useful information, the paper should be improved before being accepted for publication.

  1. Introduction

Line 42. It is stated that “Ticks transmit piroplasmosis, which act as vectors for infecting other animals”. Authors are requested to consider using instead “Ticks transmit the pathogens or etiologic agents that cause piroplasmosis…”

→ Thank you for your suggestion. It was changed as suggested at lines 41-42.

Line 46. Can use “Engorged females” instead of “They”.

→ Thank you! It was changed as suggested at line 46.

Line 50. Can use “they turn themselves as ectoparasites…” instead of “parasites”

→ This sentence was deleted during the revision.

Line 59. The sentence “They can also transmit acquired diseases to new hosts when they feed” can be better read using “They can transmit the acquired pathogen to new hosts when they feed upon”.

→ Thank you! It was changed as suggested at lines 55-56.

Line 69. Please revise the statement “The latter stages represent the differences between Babesia and Theileria parasites”, and specify if the differences are in terms of morphology or biological function in the life cycle of each parasite genus.

→ The sentence has been modified by describing the characteristics identified in Theileria life cycles at lines 65-70.

Line 72. Revise sentence “whereas Theileria do not infect erythrocytes but enter lymphocytes and develop into schizonts.”. Can use “…but infect first the lymphocytes (or monocytes, depending upon the Theileria sp) and develop into schizonts, from which erythrocyte-infective forms are released.

→ Thank you! It was changed as suggested at lines 72-74.

Line 82. Please, complete the sentence “…plasmids based on homology.” Use “…based on DNA sequence homology analysis.”

→ It was changed as suggested at line 83.

Lines 90 and 98. Use “species” instead of “spp”

→ It was changed as suggested at lines 92 and 100.

Lines 102. Authors should carefully check the statement “ When humans are affected with theileriosis…,”. Authors need to be careful about this statement. It has not been described so far that humans can be infected with Theileria spp. References 23 and 24, refer to human babesiosis.

→ I appreciate your critical comment. This sentence was deleted during revision.

Line 104.   The citation to reference number 22 is missing, it should go before these [23,24].

→ The order of missing references has been corrected. I am sorry for the mistake.

Lines 122-123. It is stated that “In Brazil, the occurrence of Theileria spp. in species of deer was reported [32,33].  Check reference 32. The paper refers to detection of Babesia and Theileria in roe deer in Spain.

→ I appreciate your critical comment. The mistake was corrected at lines 121-124.

Line 145. Rephrase statement “…(including Theileria) that can infect humans and livestock”. See the comment above on Theileria infecting humans….

The 'human' has been deleted at line 146.

  1. Materials and Methods

Line 151. Authors are asked to describe the type of study carried out (purposive), how sample size was estimated, what the estimated water deer population in Korea, etc...

→ It is estimated that here are about 700,000 water deer in Korea. The samples used in this study were taken from bodies of deer that died in a road accident. While not enough, they were obtained with the utmost effort we could secure. This is described at lines 152-154.

Line 193. Please indicate if all the small-size amplicons (Theileria sp) or any other from the larger Piroplasmid amplicon(s) were sent out for DNA sequencing

→ Thank you for your suggestion. PCR-positive samples for all PCRs mentioned in the text were sent for sequencing. This was added at lines 197-198.

  1. Discussion

Lines 272-278. Authors compare the results in terms of the infection rates obtained in their study carried out on water deer samples with “Previous studies on tick-borne diseases in Korea” and cite them in references [52] and [54]. However, these studies refer to canine pathogens (Anaplasma phagocytophilum, Hepatozoon canis, and Mycoplasma haemocanis) and cattle pathogen (Theileria sergenti) samples, respectively. Authors should indicate the relevance of such a comparison. Do water deer comingle with dogs and cattle? Do they share the same ticks as vectors??

→ I appreciate your critical comment. It was deleted during the revision.

Line 294. Use “species” instead of “spp”

→ It was changed as suggested at line 291.

Lines 295-296. Delete repeated species names “T. luwenshuni, T. lestoquardi, and T. uilenbergi”… [56]. Check for reference [55], It has not been cited up until this part of the discussion.

→ I appreciate your critical comment. The repetitive species names were deleted at lines 291-299.

Line 304. Use “species” instead of “spp”

→ It was changed as suggested at line 301.

Lines 321-322. It is stated that “From the two phylogenetic trees, it could be confirmed that the sequences obtained in this study may commonly be identified as those related to T. luwenshuni, which mainly infect goats, sheep, and deer”. Authors should be careful on this statement, as the full length ssuRNA gene was not sequenced. A blast homology analysis performed by the reviewer, using the OR 398798, 216 bp sequence and excluding T. luwenshuni from the search, showed 100% DNA sequence identity among the 216 bp amplicon and several Theileria spp. sequences, including Theileria ovisTheileria mutans, and Theileria spp in dogs.

→ Thank you for your critical comments! This limitation is further stated at lines 320-322 as follows: “As only a short length of sequence was obtained using 216 bp amplicon, it is necessary to analyze the full-length sequencing to identify species in the future.

Lines 328-330. The statement “These high similarities suggest that Theileria is potentially transmissible among the different species of mammals, ticks, and keds” should be revised. It could also mean that other Theileria spp are circulating in these hosts/vectors, not necessarily being T. luwenshuni.

→ It was deleted during the revision.

Lines 345-347. Authors conclude that comprehensive studies should be conducted “to confirm the prevalence between the host and the pathogen, prevent the spread of tick-borne pathogens, and prepare to prevent infection in wild animals and nearby livestock farms.”. Authors should speculate a little bit on how this can be accomplished, having various scenarios with different prevalence rates. How about enzootic stability / instability issues on livestock theileriosis?

→ The suggested content was added at lines 341-342 as follows: “Because enzootic stability/instability cannot be completely excluded in livestock theil-eriosis, differences in prevalence may appear in various ways.”

Comments on the Quality of English Language

To my opinion, the English language requires minor editing to understand the manuscript.

→ The revision has been got Language editing from a company, Enago. The certificate was attached.

Reviewer 2 Report

Comments and Suggestions for Authors

The manuscript I received for review "Molecular detection and genotyping of Theileria spp. in deer (Cervidae) in Korea" is an interesting work, but requires many corrections before it can be published. 

L30 – no infections instead of 0% infection rates 

I strongly recommend rewriting Introduction section, because it contains many inaccuracies and information unrelated to the paper, the aim of the study, the title, etc. The only one pathogen you detected was Therleria, Babesia has not been discovered, so there is no reason to describe in such detail a species that the study does not concern, the only (correct) use of Babesia was to include it as an outgroup in the phylogenetic tree (although there will also be comments below). Please rewrite 1st sentence (L37-38) e.g. Therleria (next to Babesia or like Babesia) belong to…. please just focus on Therleria spp. 

L48-52 – I recommend removing general information about ticks, the paper does not concern ticks and their developmental stages, but tick-borne pathogens 

L55 - the most significant (3rd - superlative form of an adjective) 

L57-59 – please rewrite this sentence, it sounds childish  

L60-73 – I recommend removing this part of the introduction  

L74 – remove was 

L75 – it should be: by a tick vector 

L98 – species instead of spp. 

L105 – rewrite this sentence, e.g. diagnosis usually based on..  

L119 – USA 

Add the Latin names of all organisms (mostly animals) given in the work (e.g. red fox, roe deer, dairy cattle etc.), you only include them sometimes (mostly discussion section) and usually not when you use the name for the first time, as it should be. 

L137 – a study – what study?? 

Blood sample collection - I'm not familiar with Korean law, but procedures such as blood collection require the consent of the Ethics Committee, which I don't see here. How exactly was the blood obtained, in accordance with what permits, according to what standards, and who performed it? Was the blood taken from live animals? Please specify here the species of deer from which blood samples were taken, this is not mentioned until the discussion where only Korean water deer is given and it is not clear whether this was the only species from which samples were taken 

Why some samples\regions are unknown? What was the reason? Explain it here. 

L193 – add number of positive samples 

L215 – samples instead of cases 

Figure 2 – the phylogenetic trees seem to have been made correctly but I recommend redoing the Figure 2 phylogenetic tree using only one Babesia as an outgroup similar to Figure 3. I also suggest remove Theileria sp. from the tree and add information about country and host in your sentences in bold (like the other from Genbank). Tree branches have no value, of course they are close to each other, but what values ​​are hidden? If they are higher than 60 you can leave them and only hide the lower ones. 

In Theileria cervi is US(A) missing in description. 

Add information about your sequences like in Figure 3. 

L257 – you mentioned here for the first time the species of deer (no Latin name) used in this study and it is a bit unclear that you mentioned infection rates from other studies, rewrite this to make it clear which of these should be taken as the results of studies by other researchers from Korea e.g. add L258  “The infection rates in Korean water deer from other studies varied…” 

L278 – other instead of different 

L302 – check the Latin names of deer, e.g. Capreolus pygarus is the Siberian roe deer and the European roe deer is Capreolus caprelous. Which of them do you mean here? 

L304 - species instead of spp.  

L309 – the most 

Comments on the Quality of English Language

I am not a native speaker, but I believe that English requires some polishing, grammatical errors are common, such as the lack of the superlative form of an adjectives or poor sentence structure. I recommend carefully translating the paper. 

Author Response

Response to Reviewer 2

Comments and Suggestions for Authors

The manuscript I received for review "Molecular detection and genotyping of Theileria spp. in deer (Cervidae) in Korea" is an interesting work, but requires many corrections before it can be published. 

→ The revision has been got Language editing from a company, Enago. The certificate was attached.

L30 – no infections instead of 0% infection rates 

→ It was changed as suggested at lines 30, 215, and 282.

I strongly recommend rewriting Introduction section, because it contains many inaccuracies and information unrelated to the paper, the aim of the study, the title, etc. The only one pathogen you detected was TherleriaBabesia has not been discovered, so there is no reason to describe in such detail a species that the study does not concern, the only (correct) use of Babesia was to include it as an outgroup in the phylogenetic tree (although there will also be comments below). Please rewrite 1st sentence (L37-38) e.g. Therleria (next to Babesia or like Babesia) belong to…. please just focus on Therleria spp. 

→ I appreciate your critical comment. The revision has been done thoroughly throughout the text. Language editing was done for this revision. This revision was focused on Theileria.

L48-52 – I recommend removing general information about ticks, the paper does not concern ticks and their developmental stages, but tick-borne pathogens 

I appreciate your critical comment. This part has been deleted.

L55 - the most significant (3rd - superlative form of an adjective) 

→ It was changed at line 50.

L57-59 – please rewrite this sentence, it sounds childish 

→ This sentence has been rewritten at lines 53-56.

L60-73 – I recommend removing this part of the introduction  

→ This paragraph describes the life cycle of Theileria and has been modified to explain its differences from Babesia at lines 65-74.

L74 – remove was 

→ Removed.

L75 – it should be: by a tick vector 

→ It was changed at line 76.

L98 – species instead of spp. 

→ It was changed at line 92.

L105 – rewrite this sentence, e.g. diagnosis usually based on..  

→ It was changed at lines 104-105.

L119 – USA 

→ It was changed at line 118.

Add the Latin names of all organisms (mostly animals) given in the work (e.g. red fox, roe deer, dairy cattle etc.), you only include them sometimes (mostly discussion section) and usually not when you use the name for the first time, as it should be. 

I appreciate your critical comment. This part was changed as suggested.

L137 – a study – what study?? 

→ The reference for “a study” was mentioned in the end of the sentence as “[11]” at line 139.

Blood sample collection - I'm not familiar with Korean law, but procedures such as blood collection require the consent of the Ethics Committee, which I don't see here. How exactly was the blood obtained, in accordance with what permits, according to what standards, and who performed it? Was the blood taken from live animals? Please specify here the species of deer from which blood samples were taken, this is not mentioned until the discussion where only Korean water deer is given and it is not clear whether this was the only species from which samples were taken.  

All blood samples were taken from bodies of deer that died in a road accident. In this case, IACUC approval is not necessary. This was mentioned at lines 153-154.

Why some samples\regions are unknown? What was the reason? Explain it here.

While individual information was labeled at the same time as the sample was collected, there was cases where the label was removed or inaccurate information was left in the process of being transferred for autopsy and specimen collection. These cases were classified as unknown samples.

L193 – add number of positive samples 

→ The “13” was added at line 197.

L215 – samples instead of cases 

→ It was changed as suggested at line 220.

Figure 2 – the phylogenetic trees seem to have been made correctly but I recommend redoing the Figure 2 phylogenetic tree using only one Babesia as an outgroup similar to Figure 3. I also suggest remove Theileria sp. from the tree and add information about country and host in your sentences in bold (like the other from Genbank). Tree branches have no value, of course they are close to each other, but what values ​​are hidden? If they are higher than 60 you can leave them and only hide the lower ones. 

→ Figure 2 presents that the obtained sequences belong to Theileria genus, but not Babesia genus. In this case, piroplasm-specific primers were used to amplify piroplasm genes, including Theileria and Babesia. In addition, we used T.gondii as an outgroup. Tree branches values higher than 60 were shown in the revised figure. Information on country and host were marked bold in the tree.

In Theileria cervi is US(A) missing in description. 

→ It was corrected in the figure.

Add information about your sequences like in Figure 3. 

→ The sequences obtained in this study are indicated by a black arrow. This was described at lines 236-237 and 239-240.

L257 – you mentioned here for the first time the species of deer (no Latin name) used in this study and it is a bit unclear that you mentioned infection rates from other studies, rewrite this to make it clear which of these should be taken as the results of studies by other researchers from Korea e.g. add L258 “The infection rates in Korean water deer from other studies varied…” 

→ Korean water deer (Hydropotes inermis argyropus) was first mentioned at line 140. Thus, the Latin name was not added here. The sentence at line 258 was changed as suggested at line 262.

L278 – other instead of different 

→ It was deleted during the revision.

L302 – check the Latin names of deer, e.g. Capreolus pygarus is the Siberian roe deer and the European roe deer is Capreolus caprelous. Which of them do you mean here? 

→ I appreciate your critical comment. It is Siberian roe deer (Capreolus pygarus). It was corrected at line 298.

L304 - species instead of spp.  

→ It was changed as suggested at line 301.

L309 – the most 

It was changed as suggested at line 306.

Comments on the Quality of English Language

I am not a native speaker, but I believe that English requires some polishing, grammatical errors are common, such as the lack of the superlative form of an adjectives or poor sentence structure. I recommend carefully translating the paper.

→ This revision has been got another language editing from Enago.

Reviewer 3 Report

Comments and Suggestions for Authors

The study is relevant to the field and the authors have worked very well to show these results. The manuscript was very elegantly written and carefully worded, and the figures were very well drawn. The analyses were carried out carefully, showing reliable results. 

Author Response

Response to Reviewer 3

Comments and Suggestions for Authors

The study is relevant to the field and the authors have worked very well to show these results. The manuscript was very elegantly written and carefully worded, and the figures were very well drawn. The analyses were carried out carefully, showing reliable results. 

→ I appreciate your positive comments. The MS was carefully revised further.

Round 2

Reviewer 1 Report

Comments and Suggestions for Authors

Authors have diligently answered all questions posted by the reviewer to  version 1 of the manuscript

Reviewer 2 Report

Comments and Suggestions for Authors

The improvement of the manuscript after review was carried out very thoroughly, the Authors corrected any shortcomings and took into account all comments.

The article is suitable for publication.

Please consider some small suggestions:

L 37 - Please choose one of the options: next to Babesia or like Babesia

Figure 3 – I suggest adding a tree branch value higher than 60 as in Figure 2.